# Enhancement of Lignin Extraction of Poplar by Treatment of Deep Eutectic Solvent with Low Halogen Content

**DOI:** 10.3390/polym12071599

**Published:** 2020-07-18

**Authors:** Jinke Liu, Letian Qi, Guihua Yang, Yu Xue, Ming He, Lucian A. Lucia, Jiachuan Chen

**Affiliations:** 1State Key Laboratory of Biobased Material and Green Papermaking, Qilu University of Technology, Shandong Academy of Sciences, Jinan 250353, China; ljk810319@gmail.com (J.L.); lqi01@qlu.edu.cn (L.Q.); xueyu@qlu.edu.cn (Y.X.); heming8916@qlu.edu.cn (M.H.); lucian.lucia@ncsu.edu (L.A.L.); chenjc@qlu.edu.cn (J.C.); 2Department of Forest Biomaterials, North Carolina State University, Box 8005, Raleigh, NC 27695-8005, USA

**Keywords:** lignin extraction, deep eutectic solvent, low halogen content, poplar, treatment

## Abstract

A novel choline-based deep eutectic solvent (DES) with low halogen content—namely choline lactate-lactic acid (CLL)—was synthesized by replacing the chloride anion with lactate anion in choline chloride-lactic acid (CCL). CLL and CCL treatments were conducted at 140 °C for 12 h with hydrogen bond acceptor/hydrogen bond donor =1/10, thereafter composition analysis and characterizations of the lignin extracted by DES treatment (DES lignin) and the solid residue were carried out. The proposed low halogen content DES presented an improved lignin extraction efficiency. The CLL treatment extracted 90.13% of initial lignin from poplar, while CCL extracted 86.02%. In addition, the CLL treatment also provided DES lignin with an improved purity (91.17%), lower molecular weight (Mw/Mn=1805/971 g/mol) and more concentrated distribution (polydispersity index=1.86). The efficient lignin extraction was mainly ascribed to the cleavage of β-O-4 bonds in lignin macromolecule, especially in the guaiacyl units, thereby breaking them into smaller molecules, facilitating the lignin extraction. The replacement of chloride anion allowed CLL acting as a more efficient DES to interact with lignin macromolecules, thus providing lignin with higher uniformity and suitable molecular weight. The low halogen content DES system proposed in present work could benefit the fractionation of biomass, improve the valorization of lignin compounds and facilitate industrial process in the downstream.

## 1. Introduction

Lignocellulosic biomass is a promising carbon-based natural alternative to fossil fuel resources due to its abundant availability, low cost and renewable feature [1]. The utilization and conversion of this sustainable biomass attracted the interest from both academic and industry [2]. Lignocelluloses are mainly composed by cellulose (40%), hemicellulose (25%) and lignin (25%). Among them the cellulose and hemicellulose are sugar polymers, whereas lignin is highly crosslinked and branched amorphous polymer [3]. The cellulose and hemicellulose can be easily converted into sugars and biofuels through the hydrolysis and fermentation process, which have been widely reported [4,5]. During the conversion process state above, lignin is considered as an inhibitor during the conversion of carbohydrates for its massive recalcitrant structure [6]. While, from the other point of view, lignin is also the most abundant renewable aromatic polymer on earth [7] and presents in all higher plants as an integral part of the plant fiber cell wall [7,8]. Thereby, the investigation of lignin extraction benefits both the valorization of lignin and the processing of cellulose and hemicellulose products. However, due to the rigid structure of plant fibers and complex crosslinks between the components [9], the lignin cannot be separated directly and efficiently without significant modification of its native structure [10]. Pretreatments are commonly used, among which the extraction of lignin with ordinary solvents have been investigated extensively and reviewed in detail elsewhere [11,12,13].

The recent discovery and development of deep eutectic solvent (DES) as an effectively solvent for the lignin extraction provided a new insight through this field [14]. The related DES was first, induced by Abbott [15] as a sustainable, biodegradable and designable solvent with great potential for varies applications [16,17,18]. In addition to its tunable and environmentally friendly feature, their strong hydrogen bonding (HB) ability presents unique solubility to the biomass, which could facilitate the selective extraction of lignin from lignocellulosic biomass [7,19]. Effective DES structures were reported being composed by tetraalkylammonium-based hydrogen bond acceptors (HBA), mixed with various hydrogen bond donors (HBD), including alcohols, carboxylic acids and amino acids. It is believed that the interactions between HBA and HBD reduce the electrostatic force between the cation and anion. Thereby, the formation of HBs in DES mixtures could reduce the freezing point of the mixture and facilitate the selective extraction of lignin compounds. The literatures [20,21,22] clearly demonstrated that choline-based DES could efficiently extract lignin from lignocellulosic, while choline chloride-lactic acid is one of the most efficient DES to extract lignin with high yield and high purity. The depolymerization mechanism of lignin was reported that DES selectively cleaved the ether linkages in wood fibers and facilitated the lignin extraction from lignocellulosic biomass.

Choline chloride ([Ch][Cl]) and its alternatives were also applied in the fractionating of lignocellulosic biomass. Their solvation chemistry of was mainly determined by their anion while the cation having a secondary effect [23]. Choline carboxylic acid [24,25] and choline amino acid [26,27,28,29,30] were synthesized and reported as effective solvents for lignocellulosic biomass pretreatment. In addition, these choline-based ionic liquids also presented low toxicity and good biodegradability [25,31]. These choline-based ionic liquids (ILs) could selectively dissolve and extract lignin. In addition, the removal of chloride anion in choline-based ILs could avoid the corrosion of process equipment [32]. Replacing the chloride anion in choline chloride-lactic acid DES with carboxylate could improve the selectivity during the lignin extraction and prevent the corrosion of process equipment.

In the present work, a novel choline-based DES—namely, choline lactate–lactic acid (CLL)—was synthesized by replacing the chloride anion in choline chloride-lactic acid (CCL). Both CLL and CCL were used to selectively separate lignin from poplar for evaluating the extraction efficiency of CLL and CCL. The purity and structural characterization of lignin were analyzed by gel permeation chromatography (GPC), Fourier-transform infrared (FT-IR) and heteronuclear single quantum coherence nuclear magnetic resonance (HSQC NMR).

## 2. Materials and Methods

### 2.1. Materials

Poplar chips were obtained from Sun Paper Co., Ltd. (Jining, China). Wood chips were selected with uniform size and grinded in a star mill. The mesh was sifted through 40–60 sieve to obtain the wood powder. Subsequently, the wood powder was extracted with benzene–alcohol solution for 8 h. The dry wood powder after the extraction was used as the lignocellulose raw material for this experiment and its chemical composition was measured. The raw hardwood sample (poplar) contains 45.46% glucose, 14.08% xylose, 1.71% mannose, 0.41% galactose, 0.21% arabinose and 25.62% lignin.

### 2.2. Chemicals

Choline chloride ([Ch][Cl]), ethanol, benzene, pyridine acetic anhydride and diethyl ether were purchased from Sinopharm Chemical Reagent Co., Ltd. (Shanghai, China). lactic acid and potassium bromide (KBr) were from Tianjin Kemiou Chemical Reagent Co., Ltd. (Tianjin, China); tetrahydrofuran (THF) and potassium hydroxide (KOH) were from Tianjin HengXing Chemical Reagent Co., Ltd. (Tianjin, China); dimethyl sulfoxide (DMSO-d_6_) were from Shanghai Macklin Biochemical Technology Co., Ltd. (Shanghai, China).

### 2.3. Preparation of DES

The choline lactate ([Ch][Lac]) was synthesized according to the literature method [33]. [Ch][Cl] and sodium hydroxide were mixed in ethanol. The mixture was kept at in 60 °C water bath for 4 h. Subsequently, the supernatant of mixture was filtered and added with lactic acid. The solvent in the supernatant was then removed by rotary evaporation under reduced pressure to obtain [Ch][Lac].

Choline chloride–lactic acid (CCL) and choline lactate-lactic acid (CLL) used in this work were synthesized by mixing lactic acid with choline chloride and choline lactate, respectively. The DES was synthesized with seven different HBA to HBD ratios (molar ratio 1/2, 1/4, 1/6, 1/8, 1/10, 1/12, 1/14).The mixture was stirred at 65 °C for 2 h to generate a well-mixed, homogeneous and transparent liquid. The obtained CCL and CLL were cooled down to room temperature in a desiccator for the further application.

The general physical parameters, including melting point (T_m_), pH, viscosity of CCL and CLL are listed in Table 1.

### 2.4. DES Treatment

DES treatment for lignin extraction was processed according to the literature method [7], where 1.5 g over-dry biomass and 45 g DES were mixed in a 100 mL pressure vessel, which was incubated at 80–160 °C for 4–14 h. The reaction was terminated by adding large amount of ethanol. The solid residue was separated by filtration, washed by ethanol and water, then dried in oven at 105 °C. Deionized water (DI water) was added into the filtrate to precipitate the extracted lignin by DES treatment (DES lignin). Lignin was collected by centrifugation and washed with additional water/ethanol mixture for purification. Composition analyses of solid residue and DES lignin were conducted following the standard NREL procedures [34]. The process of DES treatment for lignin extraction is shown in Figure 1.

The HBA/HBD ratio, process temperature and duration for lignin extraction conditions were optimized by experiments. The optimized DES treating conditions were obtained with HBA/HBD of 1/10, operating at 140 °C for 12 h. The optimal conditions were selected for further investigation in this work, in which every sample was tested at least three times and the average value was taken and reported.

The recovery of DES was carried out following literature method [35]. Briefly, the effluent generated by water and/or ethanol washing was collected and filtered to remove insoluble content. The filtrate was collected and dehydrated under reduced pressure. Repeat the filtration and dehydration process until the filtrate became a clear solution. Further dehydration was performed under reduced pressure with Schlenk line at 70 °C for 24 h. The structure and purity of the recycled DES were checked by NMR.

### 2.5. Preparation of Milled Wood Lignin

Milled wood lignin (MWL) was isolated from the poplar wood species according to the procedure reported by Björkman [36]. Briefly, poplar powder was put into a ball mill for 72 h at room temperature. Subsequently, the powder was extracted with a dioxane-water mixture and centrifuged to obtain a supernatant. The supernatant was put onto rotary evaporator to remove solvents and then freeze-dried to obtain the MWL product.

### 2.6. Analysis of Lignin Characteristics

#### 2.6.1. FT-IR Analysis

FT-IR (VERTEX70, Bruker, Karlsruhe, Germany) was used to analyze lignin characteristics. The dried samples were embedded in KBr pellets at concentrations of 1 mg/100 mg KBr. All spectra were recorded in absorption band mode over the range of 4000 to 500 cm^−1^.

#### 2.6.2. Acetylation of Lignin

For the acetylation of lignin samples, 50-mg lignin sample was dissolved in 2 mL pyridine in a 20-mL glass vial, followed by the addition of acetic anhydride (2 mL). The mixture was sealed and allowed to react at room temperature for 24 h. The acetylated lignin was precipitated in ice DI-water with constant stirring and collected by filtration. The collected acetylated lignin was washed with DI-water and vacuum-dried.

#### 2.6.3. GPC Analysis

The molecular weight distribution of lignin samples was determined by GPC (WATERS Alliance HPLC e2695, Waters, MA, USA). Approximately 2 mg of acetylated lignin sample was dissolved in tetrahydrofuran and then filtered through a 0.45-μm filter. An Agilent 1200 series high performance liquid chromatography (HPLC) equipped with an ultraviolet detector (UV) at 254 nm was used to conduct GPC analysis of lignin molecular weight distribution. Polystyrene standards with molecular weights ranging from 139 to 16,000 g/mol were used to calibrate the molecular weight based on retention time.

#### 2.6.4. HSQC NMR Analysis

NMR (AVANCE III 500 MHz, Bruker, Karlsruhe, Germany) were recorded according to literature method [7], by using a spectrometer equipped with a DCH cryoprobe. HSQC spectra was recorded at 25 °C using the Q-CAHSQC pulse program. Matrices of 2048 data points for the ^1^H-dimension (13 to −1 ppm) and 1024 data for the ^13^C-dimension (160 to 0 ppm) were collected, with the relaxation delay set at 6 s [7]. The lignin samples were dissolved in 0.5 mL of dimethylsulfoxide-d6 (DMSO-d_6_) and chemical shifts were referenced to the solvent signal (2.50/40.21 ppm).

## 3. Results and Discussion

### 3.1. Lignin Extracted by Choline-Based DES

In order to evaluate the efficiency of choline base-DES on the lignin extraction from the poplar, CCL and CLL at HBA/HBD of 1/10 (molar ratio) were prepared. The treatment was carried out at temperature of 140 °C and solid ratio of 30:1 (mass ratio) for 12 h. The sugars and lignin of samples with or without DES treatment were analyzed. The distribution of the main chemical compositions in DES solid residue and DES lignin are shown in Figure 2, in which the calculation was based on the dry initial sample. The whiskers regions in the figure indicate the compositions extracted into DES lignin, while black bars are corresponding to compositions remain in the solid residues. It could be seen that DES treatment efficiently fractionated the components of poplar samples by the lignin removal, while keeping the cellulose content in the solid residues. The raw sample contains 45.46% glucose, 14.08% xylose, 1.71% mannose and 25.62% lignin. While in the CCL treatment process, 22.04% of native lignin was extracted into CCL lignin phase and 43.37% of cellulose (glucose) was kept in the solid residues. It was intriguing to find CLL treatment presented higher fractionation efficiency, as a higher amount of lignin presented in CLL lignin (23.09%), alongside with more cellulose component (44.73%) in CLL solid residue. It should be noticed that a large amount of mannose (1.14%) was also preserved in CLL solid residue, which was not found in CCL-treated samples. Therefore, the CLL seems to be a promising solvent to achieve the selective separation of lignin from lignocellulosic biomass.

A further composition analysis as a percentage based on the initial sample component are presented in Table 2. As being widely reported [8,37], CCL in this work extracted most of the lignin (86.02%), while 95.29% of the initial glucose, 25.80% of the initial xylose and only 9.73% of the initial lignin were remained in the CCL solid residue. In comparison, the CLL treatment further improved the fractionation efficiency, as 98.40% of the initial glucose, 24.10% of the initial xylose and 67.01% of the initial mannose were remained in the CLL solid residue, while only 7.03% of initial lignin was preserved. A promising 90.13% of the native lignin was extracted into CLL lignin phase, and the extracted DES lignin could be easily collected by precipitation with water. It was clearly shown that the replacement of chloride anion with lactate anion in DES promoted the separation selectivity during the DES treatment process of lignocellulosic biomass, which elevated the lignin extraction rate from 86.02% (CCL lignin) to 90.13% (CLL lignin). More important, only trace amount of glucose (0.10%) was detected in the final CLL lignin.

All the results above clearly indicated that both CCL and CLL treatment could efficiently fractionate lignocellulosic biomass, as the glucose remained in the DES solid residue and most of lignin were extracted out as DES lignin. Between these DES treatment methods, CLL presented higher efficiency and selectivity, as better preservation of cellulose in the initial sample and higher lignin recovery rate were viewed in the CLL-treated sample. Therefore, it seems that the replacement of chloride anion in CCL with lactate anion (CLL) facilitated the fractionation of biomass, by extracting more lignin into DES lignin phase and leaving more carbohydrates in solid residue.

Indeed, even for the CLL treatment, there were still around 3% of lignin, 1% of glucose, 75% of xylose, 33% of mannose and almost all of the galactose and arabinose remained in the aqueous solutions, which may due to the thermal degradation and acid hydrolysis of hemicellulose under acid conditions [8,38,39]. However, these degraded carbohydrates presented limited impact on the DES recovery. They could be easily recovered by collecting the effluent after solvent (water and ethanol) washing, where up to 98% of the DES could be recovered.

### 3.2. Characterization of Lignin

#### 3.2.1. Lignin Purity

For the composition analysis of lignin extracted by the DES treatment, the acid insoluble lignin (AIL), acid soluble lignin (ASL) and sugar content were measured. The results are presented as a percentage based on the testing sample and shown in Table 3. Lignin extracted by both CCL and CLL treatments were of high purity (>85%), with only trace of glucose and no detectable hemicellulose residues. Lignin extracted by CCL (CCL lignin) was of 88.82% purity, with only 0.35% of glucose, while lignin extracted by CLL (CLL lignin) presented a higher purity (91.17%) with only 0.21% of glucose detected. The lignin purity analysis results further supported our hypothesis that the removal of halogen anion could improve the lignin extraction efficiency, as the choline lactate-lactic acid DES not only extracted more lignin from the biomass, but also provided CLL lignin with higher purity.

#### 3.2.2. Molecular Weight Distribution

The variation of weight average molecular weight (Mw), number average molecular weight (Mn) and the polydispersity index (PDI) during DES treatment are shown in Table 4. In comparison with the MWL lignin, DES extraction significantly reduced the Mw and Mn value of lignin. Mw and Mn of CCL lignin decreased from 10,000 and 4166 to 4416 and 2349 g/mol, respectively. At the same time, Mw and Mn of the CLL lignin could further reduce to 1805 and 971 g/mol, respectively. This indicated that both DES treatment could facile the cleavage of the lignin linkage. It should be noticed that the molecular weights of DES lignin in this work were also lower than those reported in literature extracted through organic solvents [40]. Thereby, in this work the DES lignin extracted from poplar presented unique low molecular weight, which could upgrade the value of lignin stream and facilitate further valorization. The CLL treatment provided lignin stream of a much lower molecular weight, which could attribute to a stronger or more interactions formed between the DES and lignin molecules.

In addition, it was shown in Table 4 that a more concentrated distribution of polydispersity of lignin was found with DES treatment. Both DES treatment resulted in a narrow molecular weight distribution (PDI < 2.0), while the PDI of MWL is 2.40. CLL lignin was capable to provide lignin stream with better uniformity (PDI = 1.86), which may benefit the downstream process. Thus, it verified that CLL treatments was a promising method for lignin extraction.

The results in Table 4 illustrate that the choline-based DES treatment could selectively extract lignin from poplar not only with high purity, but also with low molecular weight and good uniformity. CLL, which was obtained by replacing chloride anion in CCL with lactate anion, facilitated the cleavage of intermolecular linkage. The lignin may form a stronger interaction with CLL and degraded into smaller molecules during the DES treating process. As a result, the selectivity of lignin extraction improved and CLL lignin with higher purity, lower molecular weight and more concentrated distribution obtained. These lignins extracted with appropriate molecular weight were ideal for the downstream productions [6].

### 3.3. Structure Characterization of Lignin

#### 3.3.1. FT-IR Analysis

The FT-IR analysis of the initial sample, MWL and DES lignin are shown in Figure 3. Characteristic peaks of different bonds [8,42] and detailed signal assignments are listed in Table 5. The FT-IR analysis confirmed a high purity lignin was extracted by DES treatment, as both CLL and CCL lignin presented similar absorbance bands with those of MWL. Typical lignin benzene ring skeleton bands were found at 1604 and 1510 cm^−1^ in all lignin samples, illustrating a well-preserved lignin framework after DES treatment. Absorption bands at 1328 and 1220 cm^−1^ were corresponded to C–O stretching vibration in syringyl units, while the absorbance at 1116 cm^−1^ was related to the C–H stretching vibration. All these characteristic peaks were found in the MWL, CLL lignin and CCL lignin samples, indicating the syringyl unit was left intact after both DES extraction. The absorption at 1270 and 1032 cm^−1^ were associated with the C–O stretch and C–H bending vibration of guaiacyl unit, respectively. The feature of GS-typed lignin was demonstrated in the FT-IR spectra. It was noticed that the guaiacyl unit-related peaks presented lower intensities after both DES treatments, illustrating the cleavage of guaiacyl units occurred during the DES extraction. Therefore, the lignin depolymerization reported in GPC results may possibly attribute to the degradation of guaiacyl units, while the syringyl units and the aromatic ring remain unchanged.

In addition, the signals of carbohydrate bands at 1373 and 1162 cm^−1^ did not appear in both CCL and CLL lignin samples, while these carbohydrate characteristic peaks appeared clearly in initial sample. This confirmed a successfully selective lignin extraction by the DES treatment, which could provide valuable lignin stream with only limited hydrocarbons contaminations.

#### 3.3.2. HSQC NMR Analysis

The structural analysis and comparation of DES lignin and MWL were investigated by HSQC NMR; the results are shown in Figure 4. Assignment of the peaks in the spectra were based on previous report [37]. The peaks corresponding to β-aryl ether connections are shown for A_α_ (δ_C_/δ_H_ 4.95/69.92 ppm), A_γ_ (δ_C_/δ_H_ 3.81/61.69 ppm), A′_γ_ (δ_C_/δ_H_ 4.67/64.85 ppm), A(H/G)_β_ (δ_C_/δ_H_ 4.63/80.48 ppm) and A(S)_β_ (δ_C_/δ_H_ 4.88/78.74 ppm). β-β connections are shown for B_α_ (δ_C_/δ_H_ 4.70/85.49 ppm), B_β_ (δ_C_/δ_H_ 3.06/53.71 ppm) and B_γ_ (δ_C_/δ_H_ 3.86/71.80 ppm). β-5 connections are shown for C_α_ (δ_C_/δ_H_ 5.59/87.55 ppm), C_β_(δ_C_/δ_H_ 3.76/59.36) and C_γ_(δ_C_/δ_H_ 4.36/65.09 ppm). Moreover, methoxyl connections was shown at (δ_C_/δ_H_ 3.76/56.33 ppm). As shown in Figure 4, the MWL had abundant β-ether bond units (Figure 4d–f) as well as a small amount of resinol (Figure 4h) and phenylcoumarin (Figure 4i), which was in agreement with our previous FT-IR results, where the poplar lignin demonstrated typical GS-typed lignin.

It can be seen from Figure 4b,c that the absence of major contours corresponding to the non-anomeric carbohydrates (δ_C_/δ_H_ 65–85/2.5–4.5 ppm) in both CCL and CLL were noticed, which verified the excellent lignin extraction selectivity. What is more, after both DES treatment β-ether bond (A_α_ and A_γ_) was kept, however β-O-4 ether linkages connected to guaiacyl (Figure 4e) and syringyl (Figure 4f) could not be detected in the HSQC spectrum (blue cycle), which indicated that a cleavage of β-O-4 ether bond between the guaiacyl and syringyl structural units occurred during the DES treatment. It was reported that lignin degradation in DES was similar to the typical lignin acidolysis, during which Hibbert′s ketone was generated as intermediate [41,43,44]. As shown in Figure 4, the presence of Hibbert′s ketone (Figure 4k), 4.21/67.5 ppm) was detected in both CCL and CLL lignin samples, which verified the cleavage of β-O-4 bonds. Therefore, the HSQC analysis confirmed that DES treatment in this work facilitated the cleavage of β-O-4 bonds in lignin macromolecules and promoted the lignin extraction.

It could also be seen that after both DES pretreatment, most of the C–C bonds were retained and remained as the main interunit bonds in lignin samples, while the presence of ether bonds significantly reduced. Hence, ether bonds are more vulnerable to damage than carbon–carbon bonds [7], the decrease of ether bonds in DES lignin could provide lignin stream with an improved polymer stability and utility value.

The characterizations of CLL lignin and CCL lignin interestingly presented similar composition and structure, which indicated both DES interacted with lignin macromolecule in the same mechanism of action. It was proposed in this work that DES treatment induced the cleavage of the β-O-4 bonds in lignin, thereby degrading the lignin macromolecule into smaller molecules, facilitating the fractionation of lignocellulosic biomass. A possible mechanism was that the replacement of chloride anion with lactate anion in DES system allowing CLL to present stronger or more interactions with lignin during DES treating process. A typical reaction for the cleavage of lignin β-O-4 bonds is shown in Figure 5, in which the formation of protonated intermediates was reported [45]. Different nucleophilicities of the anion in DES altered the cation–anion interaction, consequently influenced the solvation of the protonated intermediates [46]. In this case, chloride anion was of strong electronegativity, thereby presenting strong cation–anion interaction and a weak interaction. In contrast, lactate anion with less electronegativity presented weak cation–anion association, but with strong interaction. This allowed CLL in the mixture to help with the stabilization of the protonated intermediates by forming hydrogen bonds. Therefore, the CLL could serve as a better catalyst during the cleavage of β-O-4 bonds in lignin degradation, promoting the selective extraction of lignin compounds from lignocellulosic biomass. This could contribute to the production of lignin streams with high yield, high purity, low molecular weight and good uniformity after CLL treatment, which were also more suitable for the downstream production of biofuels and functional materials.

## 4. Conclusions

CLL as a deep eutectic solvent (DES) with low halogen content was applied to enhance the lignin extraction of poplar. The replacement of chloride anion with lactate anion from CCL can reduce the potential corrosion to the equipment and also allowed CLL to present a significantly high efficiency in fractionating poplar biomass. A better preservation of glucan in the DES solid residue and a higher extraction yield of lignin in the DES lignin were obtained in the CLL treatment process. In addition, DES lignin with higher purity, lower molecular weight and more concentrated distribution were also achieved in the CLL treatment. The choline-based DES in this work could cleavage the β-O-4 bonds of guaiacyl units in lignin macromolecule, thus, breaking them into smaller molecules, as a result facilitating the lignin extraction process. The removal of chloride anion allowed the CLL acted as a more efficient DES interacting with lignin macromolecules, hence providing CLL lignin with higher uniformity and suitable molecular weight. The low halogen content DES system proposed in present work could improve the fractionation of lignocellulosic biomass, providing lignin stream as promising feedstock for downstream process, thereby benefiting the future large-scale industrial process.

## Figures and Tables

**Figure 1 polymers-12-01599-f001:**
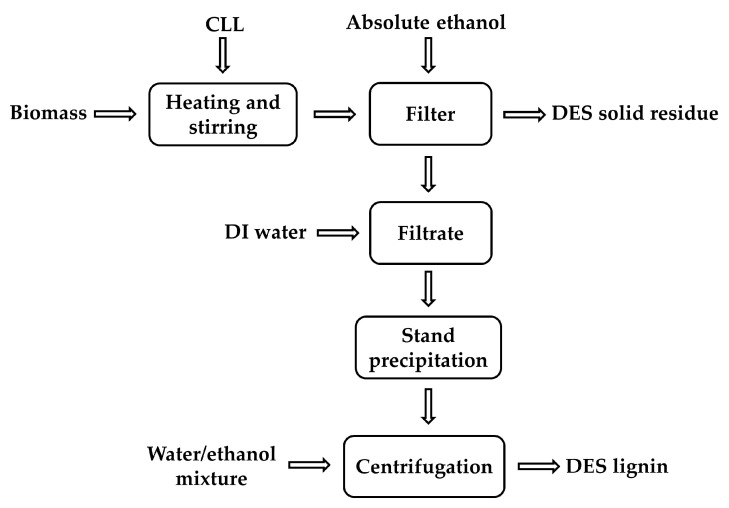
The process of lignin extraction from lignocellulosic biomass by deep eutectic solvent (DES) treatment.

**Figure 2 polymers-12-01599-f002:**
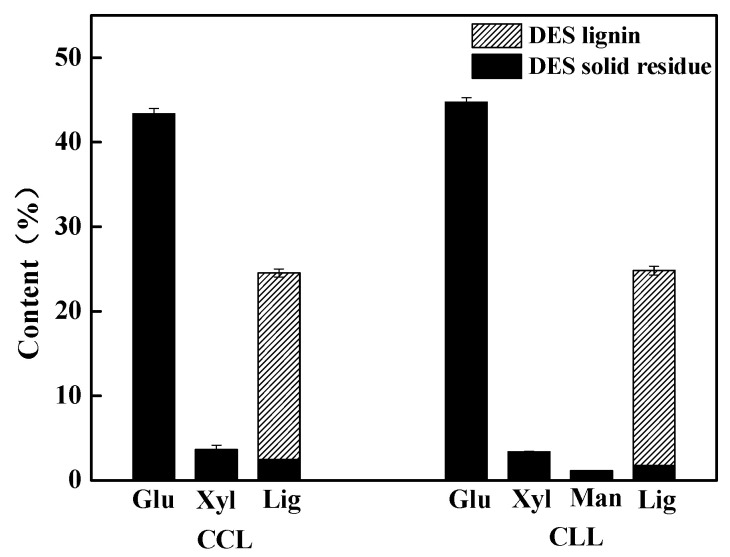
Composition analysis of samples in DES-treated solid residue and DES lignin at 140 °C for 12 h. (Based on over-dry basis of the initial sample).

**Figure 3 polymers-12-01599-f003:**
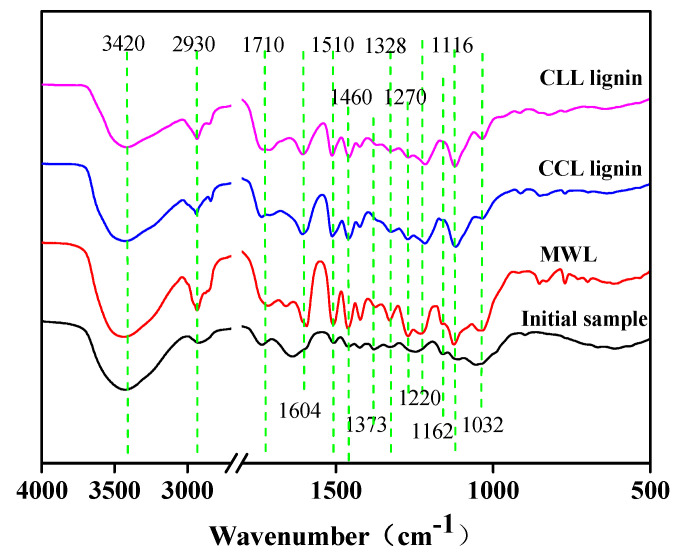
FT-IR analysis of the initial sample milled wood lignin (MWL), CLL lignin and CCL lignin.

**Figure 4 polymers-12-01599-f004:**
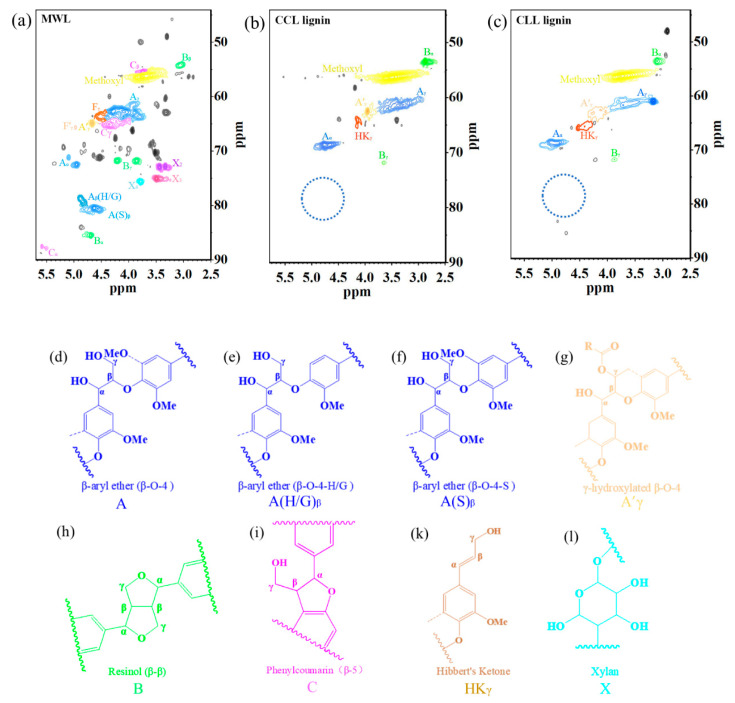
HSQC NMR analysis of MWL and DES lignin. (**a**) milled wood lignin; (**b**) lignin extracted by choline chloride-lactic acid; (**c**) lignin extracted by choline chloride-lactic acid; (**d**) β-O-4 aryl ether bond; (**e**) β-O-4-H/G aryl ether bond; (**f**) β-O-4-S aryl ether bond; (**g**) hydroxylated β-O-4; (**h**) resibol; (**i**) Phenylcomarin; (**k**) Hibbert′s ketone; (**l**) xylan. The abbreviation of A, A(H/G)β, A(S)β, A′γ, B, C, HKγ and X listed in Figure 4d–l are used to indicate the different types of lignin structure shown in Figure 4a–c.

**Figure 5 polymers-12-01599-f005:**
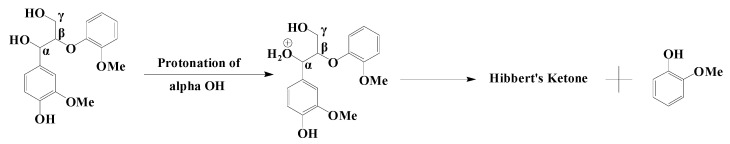
Mechanism of β-O-4 ether link cleavage of lignin compounds [45].

**Table 1 polymers-12-01599-t001:** General physical parameters of novel choline lactate–lactic acid (CLL) and choline chloride-lactic acid (CCL).

DES	T_m_ (°C)	Viscosity (Pa·s)	Density (g/cm^3^)	pH
CCL	18.34 ± 0.34	0.87	1.2022	0.91
CLL	18.14 ± 0.39	1.38	1.2078	2.75

**Table 2 polymers-12-01599-t002:** Chemical compositions of DES solid residues and DES lignin (as a percentage based on the initial sample component).

	Glucose	Xylose	Mannose	Galactose	Arabinose	Lignin
CCL solid residue	95.29 ± 3.42	25.80 ± 1.11	–	–	–	9.73 ± 0.78
CCL lignin	0.17 ± 0.1	–	–	–	–	86.02 ± 1.58
CLL solid residue	98.40 ± 0.42	24.10 ± 1.15	67.01 ± 2.21	–	–	7.03 ± 0.58
CLL lignin	0.10 ± 0.1	–	–	–	–	90.13 ± 2.63

**Table 3 polymers-12-01599-t003:** Purity of the DES lignin extracted from poplar as a percentage based on the testing sample.

	Glucose	Xylose	Mannose	Galactose	Arabinose	ASL	AIL	Lignin
Poplar	45.46 ± 1.21	14.08 ± 0.37	1.71 ± 0.19	0.41 ± 0.10	0.21 ± 0.10	5.05	20.57	25.62 ± 0.32
CCL lignin	0.35 ± 0.10	–	–	–	–	2.30	86.52	88.82 ± 1.25
CLL lignin	0.21 ± 0.10	–	–	–	–	1.92	89.25	91.17 ± 2.15

ASL—acid soluble lignin; AIL—acid insoluble lignin.

**Table 4 polymers-12-01599-t004:** Average molecular weights and polydispersity indices of MWL and DES lignin.

	Mw (g/mol)	Mn (g/mol)	PDI
Lit. MWL *	10,000	4166	2.40
MWL	6374	5542	1.15
CCL lignin	4416	2349	1.88
CLL lignin	1805	971	1.86

* Literature MWL data taken from Li, N. et al. 2018 [41].

**Table 5 polymers-12-01599-t005:** Assignment of FT-IR spectra of the initial sample, MWL, CLL lignin and CCL lignin.

Wavenumbers (cm^−1^)	Assignment (Bond)	Initial Sample	CLL Lignin	CCL Lignin	MWL
3420	O–H stretching vibration	√	√	√	√
2930	C–H stretching vibration in methyl	√	√	√	√
1710	C=O stretching vibration	√	√	√	√
1604,1510	Aromatic ring skeleton vibration	√	√	√	√
1460	C–H deformation vibration in -CH_2_-	√	√	√	√
1373	C–H bending vibration of aliphatic compounds in carbohydrate	√	×	×	×
1328,1220	C–O stretching vibration of syringyl units	√	√	√	√
1270	C–O stretching vibration of guaiacyl units	√	√	√	√
1162	C–O–C symmetrical stretching vibration in carbohydrate	√	×	×	×
1116	C–H stretching vibration of syringyl units	√	√	√	√
1032	C–H bending vibration of guaiacyl units	√	√	√	√

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
