# Peer review of "Enhancement of Lignin Extraction of Poplar by Treatment of Deep Eutectic Solvent with Low Halogen Content"

_polymers, 2020, doi:10.3390/polym12071599_

Round 1

Reviewer 1 Report

The work describes an improved lignin extraction procedure using a low halogen content DES. Generally, the paper shows correctly the data. The experimental work is also quite systematic and thorough.
I have a few issues to consider prior to publications.

-The choice of hydrogen bond acceptor to hydrogen bond donor ratio (HBA/HBD) of 1/10 (molar ratio) should be explained.

-General physical parameters (ie melting point) of novel CLL respect to CCL form should be added

-The eventually recycling aspect of CLL (recovery and reuse) after lignin extraction can be evaluated

-Suggestion: The structural analysis can be improved adding a 31P-NMR analysis in order to study the different -OH amount between CCL and CLL lignins.

Author Response

Dear Reviewer,

Thank you so much for your  constructive comments on our manuscript  submitted to Polymers. Manuscript have been revised according to your comments. All changes have been done in the “Revised Verersion” of the manuscript and all the changes and modifications are highlighted in yellow color. Please find a list of point-by-point responses to your comments attached below this letter, and the uploaded revised verersion of polymer-853540.

Thank you very much for your attention and consideration.

Sincerely regards,

Guihua Yang

Responses to Reviewer's Comments

Question 1: The choice of hydrogen bond acceptor to hydrogen bond donor ratio (HBA/HBD) of 1/10 (molar ratio) should be explained.

Response 1: Thanks! The DES was synthesized with seven different HBA to HBD ratios (molar ratio 1/2, 1/4, 1/6, 1/8, 1/10, 1/12, 1/14). The HBA/HBD ratio, process temperature and duration for lignin extraction conditions were optimized by experiments. The optimized DES treatment condition was obtained with HBA/HBD of 1/10, operating at 140 °C and 12 h. This optimized operating condition was selected for further investigation in this work, in which every sample was tested at least three times and the average value was taken and reported. Please find the detailed modification in revised manuscript in Page 3 Line 103-105 and 121-125.

Question 2: General physical parameters (ie melting point) of novel CLL respect to CCL form should be added

Response 2: Thank you for the suggestion! The general physical parameters of DES, such as Tm, pH, viscosity and density were added, the changes could be found in Page 3 Line 108-109 of the revised manuscript. Detailed results were added in Table 1.

Question 3: The eventually recycling aspect of CLL (recovery and reuse) after lignin extraction can be evaluated.

Response 3: Thanks! The recovery of DES was evaluated, where upto 98% of DES could be recovered. The recovery of DES was carried out following literature method [35]. Briefly, the effluent generated by water and/or ethanol washing was collected and filtered to remove insoluble content. The filtrate was collected and dehydrated under reduced pressure. Repeat the filtration and dehydration process until the filtrate became a clear solution. Further dehydration was performed under reduced pressure with Schlenk line at 70 °C for 24 h. The structure and purity of the recycled DES was checked by NMR. These changes could be found in Page 4 Line 128-133.

The degraded carbohydrates presented only limited impact on the recovery of DES. The DES could be easily recovered by collecting the effluent after solvent (water and ethanol) washing, where upto 98% of the them could be recovered. These changes could be found in Page 6 Line 210-212.

Suggestion: 4: The structural analysis can be improved adding a 31P-NMR analysis in order to study the different -OH amount between CCL and CLL lignins.

Response 4: Thanks for your great suggestion! Further 31P-NMR analysis would be very helpful to obtain different -OH amount between CCL and CLL lignin. This will definitely something that we are planning to test in the future work.

Reviewer 2 Report

This paper evaluated the effect of the new deep eutectic solvents (DES) on the lignin extraction. Application of DES to extraction of natural polymers is well known. Authors developed a new DES formula based on the on the choline lactate. The final DES mixture were: chloride-latic acid and choline lactate-lactic acid.

The final DES solvents were mixtures: chloride-latic acid and choline lactate-lactic acid. It is very well known that use natural DES is one of the effective and not complicated method to separation of high quality lignin. One of the DES solvent is mixture of lactic acid/choline chloride presented in some papers ( eg.  https://doi.org/10.1007/s11356-015-4780-4).

So I have the basic remark about of novelty of presented research. In this manuscript Authors need to highlight the novelty of research and importance of results which can be obtained.   Specially when used similiar solvents were presented in earlier publications.    

What Statistical Analysis methods were used to estimation of obtained values in the experiment?

Detail comments:

Line 132: I suggest add information about the type of extractor used in the experiment.

Lines: 185-188: What is the reliability of the tests carried out if the number of repetitions is not given?

Figure 2. Whiskers seen on the chart are not described in the text.

Conclusion: reviewed manuscript touch the relatively new area application of deep eutectic solvents to extraction of the organic materials. Unfortunately in presented paper basically lack is Statistical Analysis methods. Authors presented results of their research, but it is very difficult to interpret obtained values. I suggest to deep rebuild manuscript add the strong statistical analysis and clear information about novelty of this research

Author Response

Dear Reviewer,

Thank you so much for your constructive comments on our manuscript  submitted to Polymers. Manuscript have been revised according to your comments. All changes have been done in the “Revised Verersion” of the manuscript and all the changes and modifications are highlighted in yellow color. Please find a list of point-by-point responses to your comments attached below this letter, and the uploaded revised verersion of polymer-853540.

Thank you very much for your attention and consideration.

Sincerely regards,

Guihua Yang

Responses to Reviewer’s Comments

 Question 1: Line 132: I suggest add information about the type of extractor used in the experiment.

Response 1: Thanks for the carefully check of the manuscript! In this work DES treatment was carried out a 100 mL pressure vessel, correlated information was added in Page 3 Line 113.

 Question 2: Lines: 185-188: What is the reliability of the tests carried out if the number of repetitions is not given?

Response 2: Thank you for the remind. In this work every test was repeated at least 3 times, and the average value was reported. We noticed this is an important message that was missing in previous manuscript. Therefore, the related information was added in Page 3 Line 123-125, and the margin of error was added in Table 2 and Table 3.

Question 3: Whiskers seen on the chart are not described in the text.

Response 3: Thanks! For Figure 2, the whiskers regions indicate the compositions extracted into DES-lignin, while black bars are corresponding to compositions remain in the solid residues. It seems that the previous descriptions related to this part in the manuscript was not clear. Therefore, we reconstructed the manuscript in order to have a better presentation of our findings. Please find the modifications marked in yellow color in Section 3.1 for more details.

Question 4. Conclusion: reviewed manuscript touch the relatively new area application of deep eutectic solvents to extraction of the organic materials. Unfortunately in presented paper basically lack is Statistical Analysis methods. Authors presented results of their research, but it is very difficult to interpret obtained values. I suggest to deep rebuild manuscript add the strong statistical analysis and clear information about novelty of this research

Response 4: Thanks for the great advice! Sorry for the confusions that made in our previous manuscript. Similar to those modifications in Section 3.1, results and discussions in the following sections were also rebuilt to highlight the performance of CLL treatment. Related values should be easier to get, and information should be easier to interpret. Please find the revised Section 3. Results in Page 5-10 for more details.

Reviewer 3 Report

Lignin Extraction Eutectic 2020 China

The manuscript provides new information on extraction of lignin that can have impact on sustainability. Another positive aspect of the paper is reporting a novel choline-based DES, namely, choline lactate-lactic acid (CLL). The manuscript can be publishable after following changes have been made.

  • In general there are many typos and in many places the English writing is very poor. In the following I mention some examples but the authors should rewrite most of the paper with a better attention to the writing to correct many English language mistakes and typos. The English mistakes unfortunately have negative impacts on the significance of the content, and the interests of the readers.
  • All abbreviations should be explained the first time they are used.
  • Some references should be replaced with more relevant and recent references. E.g. References 1 and 2 should be replaced by ONE recent review article on application of Lignocellulosic biomass as fuels. Reference 3 should be replaced by a review article on industrial scale conversion of Lignocellulosic biomass. 5 should be replaced with an article on the hydrolysis and fermentation process of cellulose. 14 and 15 should be replaced with ONE recent review article on use of deep Eutectic Solvents in biomass processes (e.g. treatment and extraction). The list goes on so these are only a few examples. In general in many places proper references have not been cited.
  • Line 39: has been should be have been.
  • The last sentence in page 2 does not make sense due to poor English. The English should be changed.
  • Line 146: The. Also the sentence is wrong!
  • The authors should explain in the paper why the relaxation delay was set at 6 s.
  • Sections are numbered improperly! E.g. the section 3.2.2 comes after 3.3.1!
  • Lines 251 &252: Wrong sentence.
  • Section 3.3.1 is very weak. The authors assign IR peaks without any justification and without proper discussions. Those should be addressed.
  • Section 3.2.2 is also weak as the NMR spectra are not described and characterized properly.

Till all above are done the paper cannot be reviewed properly.

Author Response

Dear Reviewer,

Thank you so much for your constructive comments on our manuscript  submitted to Polymers. Manuscript have been revised according to your comments. All changes have been done in the “Revised Verersion” of the manuscript and all the changes and modifications are highlighted in yellow color. Please find a list of point-by-point responses to your comments attached below this letter, and the uploaded revised verersion of polymer-853540.

Thank you very much for your attention and consideration.

Sincerely regards,

Guihua Yang

Responses to Reviewer’s Comments

 Question 1: In general there are many typos and in many places the English writing is very poor. In the following I mention some examples but the authors should rewrite most of the paper with a better attention to the writing to correct many English language mistakes and typos. The English mistakes unfortunately have negative impacts on the significance of the content, and the interests of the readers.

Response 1: Thanks for the carefully check of the manuscript! Sorry for the inconvenience that was made by the previous manuscript. The revised manuscript was carefully checked and proofread to correct mistakes and typos. Corresponding change has been marked in yellow color in the revised manuscript.

 Question 2: All abbreviations should be explained the first time they are used.

Response 2: Thanks for the advice! Abbreviations in the manuscript were checked to ensure they were explained the first time used. Please find the modifications in Page 2 Line 55, Page 2 Line 81-82 in the revised manuscript for more details. All the modifications were marked in yellow color.

 Question 3: Some references should be replaced with more relevant and recent references. E.g. References 1 and 2 should be replaced by ONE recent review article on application of Lignocellulosic biomass as fuels. Reference 3 should be replaced by a review article on industrial scale conversion of Lignocellulosic biomass. 5 should be replaced with an article on the hydrolysis and fermentation process of cellulose. 14 and 15 should be replaced with ONE recent review article on use of deep Eutectic Solvents in biomass processes (e.g. treatment and extraction). The list goes on so these are only a few examples. In general in many places proper references have not been cited.

Response 3: Thanks for the suggestions! Reference 1 and 2 have been replaced by a Review Article on lignocellulosic biomass utilizations. Reference 3 has been replaced by a Review Article about the conversion of lignocellulosic biomass. Reference 5 has been replaced by as recent publication on hydrolysis of cellulose. Reference 14 and 15 has been replaced by a Review Article on the application of DES in lignocellulosic biomass. New reference at 14, 23, 35 and 42 were added. The changes were all marked in yellow color as shown in the revised manuscript.

Question 4: Line 39: has been should be have been.

Response 4: Thanks! The description has been corrected to “have been”. Please find the modifications in Page 1 Line 40.

 Question 5: The last sentence in page 2 does not make sense due to poor English. The English should be changed.

Response 5: Thanks for the advice! The sentence has been modified. Please find Line 80-82 in Page 2 of the revised manuscript.

Question 6: Line 146: The. Also the sentence is wrong!

Response 6: Thanks! The sentence has been modified. Please find Line 142 in Page 4 of the revised manuscript.

Question 7: The authors should explain in the paper why the relaxation delay was set at 6 s.

Response 7: Thanks! The HSQC analysis was carried out by using the same method that was reported by Carlos Alvarez-Vasco, et al.  (DOI: 10.1039/C6GC01007E). The reference was added in the revised manuscript, please find the modification in Page 5 Line 163.

 Question 8: Sections are numbered improperly! E.g. the section 3.2.2 comes after 3.3.1!

Response 8: Thanks for the carefully check of the manuscript! The Section number was corrected, that modification was shown in Page 8 Line 282.

 Question 9: Lines 251 &252: Wrong sentence.

Response 9: Thanks for the carefully check of the manuscript! The sentence has been modified. Please find the changes marked in yellow color as shown at line 259-260 of Page 7 in the revised manuscript.

Question 10: Section 3.3.1 is very weak. The authors assign IR peaks without any justification and without proper discussions. Those should be addressed.

Response 10: Thanks for the suggestion! Assignment of FT-IR peaks were referred to the work reported by Li, et al. (DOI: 10.3390/ijms18112266.) and Zhou, et al. (DOI:10.1016/j.indcrop.2020.112232), which was added in the revised manuscript. We have rewritten the corresponding section, presenting more detailed information and more discussions. The changes were marked in yellow color as shown at Line 259-277 of Page 7 and Page 8 in the revised manuscript.

Question 11: Section 3.2.2 is also weak as the NMR spectra are not described and characterized properly.

Response 11: Thanks for the suggestion! We have rewritten the HSQC section, presenting more detailed information and more discussions. The changes were marked in yellow color as shown at Line 282-309 of Page 8 and 9 in the revised manuscript.

Round 2

Reviewer 2 Report

Dear Authors, thank you for improve the manuscript, I am satisfied for taken into account of my suggestions to the reviewed study. I will recommend Editor to send the manuscript to publication.